# Identification of Performance Variables in Blind 5-A-Side Football: Physical Fitness, Physiological Responses, Technical–Tactical Actions and Recovery Variables: A Systematic Review

**DOI:** 10.3390/sports14010003

**Published:** 2026-01-01

**Authors:** Boryi A. Becerra-Patiño, Aura D. Montenegro-Bonilla, Wilder Geovanny Valencia-Sánchez, Jorge Olivares-Arancibia, Rodrigo Yáñez-Sepúlveda, José Pino-Ortega

**Affiliations:** 1Programa de Doctorado en Ciencias de la Actividad Física y del Deporte, University of Murcia, San Javier, 30720 Murcia, Spain; 2Facultad de Educación Física, Universidad Pedagógica Nacional, Bogotá 111166, Colombia; babecerrap@pedagogica.edu.co; 3Instituto Universitario de Educación Física, Universidad de Antioquia, Medellín 050010, Colombia; wilder.valencia@udea.edu.co; 4Facultade do Desporto, Universidade do Porto, 4050-313 Porto, Portugal; 5AFySE Group, Research in Physical Activity and School Health, School of Physical Education, Faculty of Education, Universidad de las Américas, Santiago 7500975, Chile; jorge.olivares.ar@gmail.com; 6Faculty Education and Social Sciences, Universidad Andres Bello, Viña del Mar 2520000, Chile; rodrigo.yanez.s@unab.cl; 7Faculty of Sport Science, University of Murcia, 30100 Murcia, Spain; josepinoortega@um.es

**Keywords:** blind soccer, physiological responses, sports technique, visual impairment

## Abstract

**Background:** Blind 5-A-side football is an intermittent sport that requires the development of specific physical, physiological, and technical–tactical variables, making the identification of recovery processes such as sleep, well-being, and athletes’ perceptions key factors in performance. However, to date, no systematic review has analyzed the scientific evidence on performance variables in players with visual impairments. **Objective**: To identify performance variables in blind 5-A-side football through the analysis of physical fitness factors, physiological demands, technical–tactical actions, and recovery variables. **Materials and Methods**: The following databases were consulted: Scopus, PubMed (Medline), Web of Science, ScienceDirect, and Google Scholar. This systematic review follows the PRISMA guidelines and those for conducting systematic reviews in sports science. The PICOS strategy was used to select and include studies. The quality of the studies was assessed methodologically using the Joanna Briggs Institute Critical Appraisal Tool. **Results**: The included studies evaluated multiple aspects of physical and physiological fitness in blind 5-A-side football, with a predominance of descriptive and observational research, although longitudinal interventions in national teams were also identified. The most studied physiological-physical variables are aerobic capacity and cardiovascular response; anthropometry and body composition; strength, power, and injury risk; external competition demands; balance; and postural control. The studies in the technical–tactical dimension focused on the effectiveness of shots on goal and on the characterization of control, dribbling, and shooting actions. The most studied recovery variable was sleep. **Conclusions**. The evidence suggests that training processes should integrate both improvements in physical fitness and physiological demands, as well as the refinement of decision-making and offensive actions. Despite advances, scientific output in this discipline remains limited, highlighting the need to promote studies with greater methodological rigor and sample diversity.

## 1. Introduction

Blind 5-A-side football is a sport for people with visual impairments [1]. Although blind 5-A-side football is classified by players’ LogMAR-based visual impairment (B1: <2.60, B2: 1.50–2.60, B3: 1–1.40) [2] a comprehensive systematic review of performance-related variables—including physical, physiological, technical, tactical, and recovery indicators—remains lacking in the literature, despite ongoing research into these requirements.

The performance requirements in 5-A-side football are determined by physical aspects related to optimal strength for accelerating (ACC), decelerating (DEC), and wrestling [3], cardiorespiratory fitness to maintain high intensity levels throughout the game [4], and speed in dribbling the ball to advance to the finishing zone [5]. These physical demands also elicit anaerobic and aerobic physiological responses [6,7] to promote recovery processes [8] and acute and chronic adaptations. These competitive demands have been evaluated based on the external load. It has been reported that the result is a determining factor in physical and physiological demands, with the losing teams showing higher loads (ACC, DEC, ACC/min, DEC/min) and the winning teams exhibiting this phenomenon in the first phase of the tournament [3]. It has also been reported that athletes perform more accelerations per minute (ACC/min) when they lose a match [9].

Technical and tactical variables are considered fundamental for adapting to the specific context and situation of the competition, which has proven to be an indicator of success, such as shots on goal [10]. Blind 5-A-side football has evolved in its technical and tactical demands, moving from a game based on dribbling and shooting on goals to a more combinative game through the dynamics of passing, controlling, and dribbling to achieve the objective of the game by kicking the ball toward the goal [10]. This precision is favored when there is greater balance [11], a quality that directly affects technical execution in the sport [12]. Likewise, 5-A-side soccer for the blind is a team sport in which physical contact is constant due to visual impairment, and the game requires high-intensity technical skills, integrating physiological systems with the main technical–tactical characteristics [13] and the expression of physical variables for passing, dribbling, and shooting. In this sport, players express different technical and tactical characteristics, as well as physical, physiological, and morphological traits that allow them to adapt to unpredictable situations [14,15], which together constitute determinants of performance [16]. Therefore, performance in blind 5-A-side soccer is considered a multifactorial, complex, and dynamic process [17,18,19] that requires the integration of physical abilities, physiological processes, and recovery processes that enhance technical skills within a framework of tactical understanding of the game.

The recovery variables most studied in blind 5-A-side football have focused on sleep characteristics and demographic variables, suggesting that training volume can affect sleep quality [20]. Similarly, overtraining-induced exhaustion appears to be a risk factor for athletes’ sleep and performance [21]. Other factors associated with player recovery include general well-being, pain perception, and spatial-temporal, mechanical, and metabolic workload [22]. In fact, reports of increased muscle pain and stress may increase as competition progresses, thereby increasing the risk of injury [22].

Athletic performance in blind 5-A-side soccer is a complex process that requires the interaction of different abilities, including aerobic-anaerobic systems, muscle strength, and balance, which, together with dribbling speed, shooting accuracy and power, and coordination, are essential for adapting to the demands of competition [23,24,25]. Blind 5-A-side soccer is an intermittent sport with specific requirements associated with high physiological, physical, technical, and tactical demands that optimize player performance in a competitive context with high levels of uncertainty [26]. Therefore, knowing which performance variables have been most studied in scientific literature could help to understand the main characteristics of athletes and the least explored variables with a view to designing an increasing ecological perspective of this sport [27].

This systematic review aims to summarize current evidence on the most significant performance variables—physical fitness, physiological responses, technical–tactical actions, and recovery factors—in blind 5-A-side football. By establishing these reference patterns, our work will inform coaches, sports professionals, researchers, and organizations, supporting further studies to deepen the understanding of performance in this discipline. Despite recognition that variables influence performance, existing research remains scarce, and no systematic review has yet addressed all aspects together within blind 5-A-side football.

## 2. Materials and Methods

### 2.1. Design

This systematic review followed the guidelines of the PRISMA (Preferred Reporting Items for Systematic Reviews and Meta-Analyses) statement [28] and the sports science guidelines for conducting systematic reviews [29]. The review protocol was registered on the International Platform of Registered Systematic Review and Meta-analysis Protocols (INPLASY) website on 31 October 2025 (INPLASY2025110014). Appendix A can be found in the Appendix A.

### 2.2. Sources of Information

The search strategies considered the following characteristics: Date: 31 August 2025.

The following databases were consulted: Scopus, PubMed (Medline), Web of Science and Science Direct. Searches were also conducted on Google Scholar to increase the identification of documents.

### 2.3. Inclusion and Exclusion Criteria

The two authors (B.A.B.-P; A.D.M.-B.) who led the information search and screening conducted the process independently to avoid bias. The aim was to identify articles that met the criteria established for this review (see Table 1). Any disagreement (5% of the total documents) regarding final inclusion or exclusion was resolved through academic debate among the research group during both the selection and inclusion phases. After identifying the selected studies in each database, the files were downloaded in CSV and Excel formats to create a single database that would consolidate all the retrieved information. The selection criteria for studies were defined as follows: authors, title, keywords, abstract, year, journal, citations received, and DOI. The selection and inclusion of studies in this review were established based on the inclusion and exclusion criteria derived from the PICOS strategy (Table 1). When the condensed database was reviewed, duplicate studies were identified, and records that did not appear in the search equation (retrieval) were retrieved.

The inclusion criteria were as follows: (i) studies published without language restrictions; (ii) original studies; (iii) quantitative studies; (iv) explicitly integrate and discuss the secondary literature. The exclusion criteria were as follows: (i) abstracts, meetings, books, reviews, letters, and editorials; (ii) articles written without academic peer review; (iii) studies without full access to the original text; (iv) gray literature; (v) systematic reviews, meta-analyses, bibliometric analyses, narrative or literary reviews.

### 2.4. Search Strategy and Data Extraction

Table 1 was compiled to contextualize the sample for each of the included studies. To design the search strategy, the P (population), I (intervention), C (comparison), O (outcomes), and S (study design) strategies were applied, as suggested by the guidelines used for this systematic review [29]. The Boolean operators “AND” and “OR” were used to group the terms. A similar procedure was followed for each database. Before the final search phrase for each database was constructed, possible combinations were tested with the following list of words: (“blind 5-A-side soccer players” [All fields]) OR (“blind players” [All fields]) OR (“athletes of 5-A-side football” [All fields]) OR (“blind soccer” [All fields]) AND (“match analysis” [All fields]) OR (“competition” [All fields]) OR (“performance” [All fields]) OR (“activity” [All fields]) OR (“physical demand” [All fields]) OR (“physiological response*” [All fields]) OR (“pace monitoring” [All fields]) OR (“wearable*” [All fields]) OR (“internal workload” [All fields]) OR (“external workload” [All fields]) OR (“physical fitness” [All fields]) OR (“technical actions” [All fields]) OR (“tactical actions” [All fields]) OR (“recovery variables” [All fields]) OR (“competition monitoring” [All fields]) OR (“visual impairment” [All fields]) AND (“blind person*”).

This search query was used to identify studies in each database. In addition, keyword vocabulary control was implemented to improve document retrieval in other languages. Searches were conducted to identify studies without other restrictions on publication date, language, or study design, provided they were quantitative and reported results associated with performance variables. When the full texts of studies identified through institutional subscriptions or open access were unavailable, attempts were made to contact the corresponding authors directly by email. When this was not possible, the ResearchGate platform was used, as suggested by other studies that have implemented this methodology [30].

### 2.5. Quality Assessment

The methodological quality from the articles included in this review was assessed using the Joanna Briggs Institute (JBI) Critical Appraisal Tools [31,32]. Each study was evaluated using the checklist appropriate to its design: analytical cross-sectional, quasi-experimental, cohort or studies reporting reliability and validity of measurement instruments. The checklists comprised 8–9 items each, scored as Yes, No, Unclear, or Not applicable. Risk of bias was classified as low, moderate, or high based on the proportion of positive (“Yes”) responses.

All 23 included articles were appraised using JBI tools. The majority (*n* = 16) were analytical cross-sectional studies, three were quasi-experimental pre-post studies, two were prospective cohort studies, and two were methodological validation studies. All included studies demonstrated acceptable internal validity (≥70% ‘Yes’ responses) and 16 were rated as low-to-moderate risk of bias while three were rated as low risk.

Overall, the methodological quality was low to moderate risk of bias. The main limitations observed were the lack of control groups, small sample sizes, and absence of adjustment for confounders. Conversely, all studies use validated measurement tools and appropriate statistical analyses. A summary of the critical appraisal scores is provided in Table 2.

## 3. Results

### 3.1. Identification and Selection of Studies

A total of 23 studies were identified. After an initial review of the final database, 40 documents were eliminated due to duplication (25 on the databases and 15 from the other methods). Documents that were not related to the topic after the title/abstract/keywords were reviewed were excluded from the databases (*n* = 263) or other methods (*n* = 102). Twenty-four screened documents were analyzed in depth through a systematic reading (Figure 1). After this analysis, 23 studies met the eligibility criteria.

### 3.2. Analysis of Studies

Table 3 details the classification of methodological procedures in studies according to physical and physiological variables. Table 4 reveals the methodological procedures for the technical–tactical variables, and Table 5 establishes the methodological guidelines for the recovery variables.

## 4. Discussion

This is the first systematic review to analyze scientific evidence on performance variables in players with visual impairments. In view of this, the outcome measures in this systematic review were those evaluated in at least 3 of the 23 articles. However, it should be noted that the methodological heterogeneity of the included studies, population differences, and contextual factors significantly influence the consistency and applicability of the findings. Therefore, the main variables analyzed in each of the performance indicators examined in this review are presented below.

These variables were in the physical/physiological dimension: aerobic capacity and cardiovascular response; anthropometric and body composition variables; strength, power, and injury risk; external competition demands; and balance and postural control. Technical–tactical dimension: Technical variables evaluate and measure instruments, samples, and units of analysis, including efficiency, temporal context, and competition phase; technical–tactical actions and opposition; Kinematics and effects of vision; reliability and validity of measurements; comparisons; trends; and integrative synthesis. Recovery dimension: Sleep and subjective recovery quality, fatigue and burnout, and biochemical and hormonal markers.

### 4.1. Physical/Physiological Dimensions

The included studies evaluated multiple aspects of physical and physiological fitness in blind 5-A-side football, with a predominance of descriptive and observational research. However, longitudinal interventions in national teams were also identified.

#### 4.1.1. Aerobic Capacity and Cardiovascular Response

In terms of aerobic capacity, reported VO_2_max values range from 44 to 52 mL·kg^−1^·min^−1^ in soccer players [46], with significant improvements after 14–16 weeks of in-season training programs [8,31]. In the Brazilian Paralympic team, a VO_2_peak of 51.8 ± 5.8 mL·kg^−1^·min^−1^, a maximum speed of 17.1 ± 1.4 km·h^−1^ and a maximum heart rate of approximately 190.4 ± 7.5 bpm were found [35].

In terms of cardiovascular response, average heart rates of 161 bpm were observed in competitions, with peaks above 180 bpm, equivalent to 85–90% of HRmax, which is associated with the internal demands of this sport at the level of conventional soccer [7]. PCA revealed that younger players have higher HRs and more ACCs, whereas goal scorers exhibit frequency peaks associated with explosive actions [47]. In addition, improvements in %HRmax at the respiratory compensation point were evident after prolonged training programs, reflecting positive adaptations in cardiorespiratory efficiency [33].

#### 4.1.2. Anthropometric and Body Composition Variables

The studies describe profiles where the average body mass is between 64.9 and 81.8 kg, the height is approximately 163.6 and 181 cm, the BMI is in the range of 22.3 and 25.6 kg·m^−2^, the lean mass is between 43.6 and 45.6 kg, and a somatotype with a predominance of mesomorphic-endomorphic mass [14], with variations depending on the position in the field [32,34]. Argentine players were found to have a predominantly mesomorphic somatotype and significant correlations between muscle mass and ball speed [40]. Similarly, another study reported no statistically significant differences in fat mass or fat-free mass between visually impaired males and females and sighted athletes (*p* > 0.05).

#### 4.1.3. Strength, Power, and Risk of Injury

The evaluations included jump, flexibility, and isokinetic tests. Interlimb asymmetries of approximately 6% in knee torque have been reported, which could increase the risk of injury [36]. In addition, significant deficits in jumping and flexibility were documented, along with a greater functional risk in players with visual impairments, as evidenced by the LESS test (*p* = 0.008) [39].

#### 4.1.4. External Demands of Competition

The use of GPS and heart rate monitoring shows that blind soccer players cover total distances of 993–1820 m per game, average speeds of ~2.0 km·h^−1^, with peaks of ~8–9 km·h^−1^, and average heart rates of ~161 bpm in real game situations [7]. PCA revealed that the number of ACCs per minute and the distance covered at high intensity are discriminating factors between winners and losers, and that goal scorers accumulate greater distances in the 21–24 km·h^−1^ speed range [9]. It has also been observed that the competition phase (e.g., playoffs) “modulates” these values, increasing physical demands [3].

#### 4.1.5. Balance and Postural Control

Postural balance assessed under static or semidynamic conditions shows that B1 players have a smaller displacement area than B2/B3 players and players in other adapted sports [37]. Compared with sighted individuals, blind soccer players show greater oscillation of the center of pressure with their eyes open; however, when their eyes are closed, their results are like those of sighted individuals, suggesting specific proprioceptive adaptations [48]. A comparative study between men and women revealed no significant differences in age or BMI, although differences in postural stability were reported according to visual condition [6].

### 4.2. Technical–Tactical Dimension

The studies in the technical–tactical dimension focused on the effectiveness of shots on goal and on the characterization of technical–tactical actions (control, dribbling, shooting) in blind 5-A-side football. Most of the studies used video analysis of official matches coded with the IOlF5C battery (a validated instrument for measuring the effectiveness and characteristics of play), supplemented by experimental laboratory studies using motion capture, EMG, and simulated vision paradigms to explore the perceptual-motor mechanisms behind performance [5,11,41,49].

#### 4.2.1. Sports Technique Variables and Measurement Instruments

Competition analyses coded contextual variables (match phase, minute, score), play variables (starting zone, type of progression: control + shot, dribble + shot, counterattack), and technical variables of the shot (leg, part of the foot: instep/toe/inside, rebound, opposition) via the IOlF5C and official video systems [47]. Laboratory studies have measured kinematic and electromyographic parameters, such as linear speed during sprints with the ball, angular speed during turns, trunk angles, and muscle activation (EMG), and have used motion capture and PCA of EMG to identify activation patterns [11,49].

#### 4.2.2. Samples and Unit of Analysis

The competition studies analyzed a considerable number of throws as a unit of analysis. For example, approximately 1497 throws were recorded at a World Championship, approximately 730 throws at the Rio 2016 Paralympic Games [41], and nearly 424 throws at the 2021 World Grand Prix [10]. Overall, some aggregate analyses between championships exceed 2000 shooting actions [42]. In contrast, experimental studies used smaller samples, including visually impaired players and sighted participants under simulated vision conditions; sample sizes varied across trials and, in some cases, were not reported in the abstracts [11,49].

#### 4.2.3. Shooting Effectiveness

In terms of effectiveness, the success rate per shot in international competitions ranges from 8% to 20%, depending on both the tournament and the operational definition of “success,” whether understood as a pure goal or a goal with a favorable rebound [5,10]. Analyses show that the probability of scoring increases significantly when the shot is taken from an offensive area near the goal and when it is struck with the instep or toe. These associations were confirmed by statistical contingency and logistic regression analyses of the studies presented above. In contrast, defensive opposition appears to be a limiting factor: shots with low opposition had a clearly higher probability of success than those taken under intense defensive pressure [41,50].

#### 4.2.4. Time Context and Competition Phase

In terms of time context, studies identify consistent patterns in which shot peaks occur in the 5–10 min and 30–35 min periods of the match, as well as a notable increase in the knockout phases, characterized by greater competitive demands [47]. Similarly, accumulated fatigue and score pressure modulate effectiveness, such that accuracy tends to decrease in the final minutes and under greater contextual pressure [41].

#### 4.2.5. Technical–Tactical Actions and Opposition

With respect to the type of technical–tactical action, plays that combined control plus shooting or dribbling plus shooting from offensive areas and with reduced opposition showed a higher probability of success than those performed from deeper areas or under high opposition. Similarly, the mechanics of the shot were decisive, as shots executed with the instep or toe had higher success rates. These results suggest that, under conditions of competitive pressure, the direction of play and the part of the foot used are critical variables for offensive performance [3,5].

#### 4.2.6. Kinematics and the Effect of Vision

Laboratory studies on Kinematics and perception provide complementary evidence. Visual restriction, whether real or simulated, significantly reduces accuracy in shots and penalties, decreases maximum sprint speed with the ball, and elicits technical modifications, with greater trunk flexion and increased muscle coactivation, which are understood as a safe strategy in the face of perceptual uncertainty [11,49]. Similarly, electromyographic analysis via PCA revealed that the first component (PC1), associated with the magnitude of muscle activation, tends to decrease in sighted players after training. In contrast, in blind players, this reduction is less pronounced. These findings suggest that motor modulation adapts differently across visual conditions.

#### 4.2.7. Reliability and Validity of the Measurements

Finally, the IOlF5C battery showed adequate content validity indices (high Aiken V values) and high internal consistency (Cronbach’s α ≈ 0.89) [9]. Similarly, the specific training of coders enables high intercoder kappa coefficients, supporting the comparability and robustness of studies that have used this instrument in the technical–tactical analysis of 5-A-side blind football [41].

#### 4.2.8. Comparisons, Trends, and Integrative Synthesis

Competition studies, which use a standard coding framework (IOlF5C), allow for the identification of reproducible patterns: (1) the starting zone and type of contact are the most robust determinants of success; (2) the opposition and phase of the match consistently modulate performance; and (3) championships differ in absolute frequencies and moments of most incredible intensity, which requires contextualization of training (e.g., adapting simulations to the temporal peaks observed in each tournament) [9]. Experimental studies provide causal support for the role of visual information in striking accuracy, but its direct transfer to competition is limited by differences in context (laboratory vs. actual match).

### 4.3. Recovery Dimension

Overall, the available evidence indicates that recovery in blind 5-A-side football depends not only on immediate physiological processes but also on psychological and behavioral factors. The prevalence of sleep disturbances (≈25–33%) is comparable to that observed in conventional athletes and is related to both training load and psychological stress. In addition, burnout emerges as a key modulator that compromises sleep quality and, consequently, overall recovery. At the physiological level, acute biochemical responses indicate the activation of fatigue and adaptation mechanisms. However, blind footballers are less likely to exhibit classic markers of muscle damage than their sighted peers.

#### 4.3.1. Sleep and Subjective Recovery Quality

The studies included in this dimension mainly evaluated sleep quality as a central indicator of recovery in blind 5-A-side football and in visually impaired athletes from different sports. Among a sample of players from the Chinese national team (*n* = 60, category B1), approximately 26% had poor sleep quality, as measured by the Pittsburgh Sleep Quality Index (PSQI ≥ 5) [20]. Similar results were found in a smaller follow-up panel (*n* = 10), where the average PSQI score was 4.4 ± 2.7, and 33% of participants showed significant alterations [21]. In a Japanese study of visually impaired athletes across different disciplines (*n* = 99), players with higher PSQI scores tended to have lower weekly training volumes and poorer regulation of sleep schedules, reinforcing the interaction between behavioral factors and recovery [44]. These findings show that sleep problems are common in adapted sports and have direct implications for competitive preparation.

#### 4.3.2. Fatigue and Burnout

A longitudinal follow-up by Li et al. [21] revealed a significant relationship between burnout and sleep quality: burnout had a delayed adverse effect on sleep quality, whereas the reverse direction was not substantial. These findings suggest that accumulated psychological exhaustion is a critical factor that can impair subjective and physiological recovery, underscoring the need to prioritize burnout prevention in the comprehensive management of blind soccer players.

#### 4.3.3. Biochemical and Hormonal Markers

In the area of physiological recovery, Gomes et al. [44] compared blind soccer players (*n* = 8) with sighted players (*n* = 15) during a maximum exercise test. Both groups showed significant increases in lactate (≈4 times above baseline values) and cortisol after exercise, confirming an acute metabolic and hormonal response to exercise. However, no relevant changes in creatine kinase (CK) or oxidative stress biomarkers were observed in blind soccer players, whereas sighted players presented significant increases in liver enzymes (ALT and AST). These results suggest that blind players may experience less cellular damage and oxidative stress after maximal exertion, possibly due to specific training adaptations or differences in the relative intensity achieved.

This evidence reinforces the need for a comprehensive approach to recovery in blind 5-A-side football, combining subjective monitoring strategies (PSQI, fatigue questionnaires), control of psychosocial factors (stress, burnout), and objective biomarkers (lactate, cortisol, HRV) [51].

### 4.4. Limitations and Future Recommendations

This systematic review has several limitations that should be considered when interpreting the results. First, although the PRISMA criteria and PICOS strategy were applied, the SportDiscus database was not included in the systematic search, which may have limited the retrieval of relevant literature in sports science. Second, the selection and methodological evaluation of the studies were conducted by only two reviewers, without the participation of a third independent evaluator, which could have introduced bias into the consensus during the inclusion and exclusion process.

Notably, the authors of studies whose full texts were unavailable were not contacted, leading to the exclusion of potentially relevant research and potentially introducing availability bias. Furthermore, the methodological heterogeneity of the included studies, both in terms of sample size and experimental design (cross-sectional, observational, or longitudinal), prevents direct comparisons or quantitative meta-analyses. In addition, most research has focused on high-performance male populations, limiting the generalizability of the findings to visually impaired soccer players at the training level or to female players. Likewise, the limited availability of databases and the scarcity of publications in other languages may have restricted the identification of relevant studies. Finally, a significant limitation is that some studies combine B1 players with B2/B3 players and sighted participants, which introduces a significant methodological bias in the interpretation of results when it comes to understanding the determinants of performance in blind 5-A-side soccer players.

The lack of experimental control over variables such as visual category (B1–B3), playing position, or stage of the season introduces potential biases in the interpretation of performance results. Another significant limitation is the lack of information on playing positions and their differences in performance variables, which prevented robust comparisons between tactical roles within the team. Overall, the heterogeneity in designs, sample sizes, and methodologies across the included studies limits quantitative comparisons and meta-analyses, thereby restricting the generalizability of the findings.

Future research should expand the study of technical–tactical variables using tracking technologies (GPS, accelerometry, computer vision, and three-dimensional kinematic analysis), as well as the integration of psychophysiological measures (stress, well-being, and sleep) in competitive contexts. It would also be relevant to explore the relationship between anthropometric indicators and neuromuscular adaptations to establish performance profiles. Longitudinal studies should also be developed to evaluate the impact of specific training programs on the effectiveness of technical–tactical actions and on injury prevention. Finally, the inclusion of women and young athletes with visual impairments should be promoted to expand the evidence base on sex, age, countries and competitive-level diversity.

### 4.5. Practical Application

The results of this review provide useful information for coaches, physical trainers, and sports scientists. However, it should be noted that these considerations should be generalized with caution, given the exploratory nature of the evidence, the heterogeneity of the samples evaluated and instruments used, and the differences in each context intervened upon. Based on the interpretation of the results obtained from the studies included and the variables analyzed in this review, training programs should focus on developing aerobic capacity, seeking to achieve maximum oxygen consumption (≈50 mL·kg^−1^·min^−1^), as well as stimulating explosive strength and postural control. For technical–tactical variables, it is suggested that exercises focus on striking with the instep or toe and in simulated situations of low opposition from offensive areas, which is likely to increase the effectiveness of shots on goal. From a recovery perspective, sleep control, subjective well-being, and pain perception could be preventive indicators of the risk of overtraining and injury. Performance analysis in blind 5-A-side soccer requires comprehensive preparation and the use of tools adapted to the needs of the athletes. This is especially important because scientific evidence should provide information that can be applied in the real context of training and sports preparation [52,53].

## 5. Conclusions

Although the available scientific evidence is not consistent, it appears that certain variables related to physical, physiological, technical–tactical, and recovery factors may influence the athletic performance of players who participate in blind 5-A-side soccer. Some of the most representative physical variables appear to be balance and postural control, while cardiorespiratory fitness seems to be one of the physiological variables that influence performance. Among the technical–tactical variables, one of the most representative is accuracy in shooting in offensive areas with minimal opposition.

Scientific output in this discipline remains limited, highlighting the need to promote studies with greater methodological rigor and sample diversity, which makes it difficult to extrapolate these findings to different contexts. Therefore, it is suggested that these findings be reviewed with caution. This review lays the foundation for future research and could serve as input to guide practical applications that seek to understand the athletic performance of blind soccer players.

## Figures and Tables

**Figure 1 sports-14-00003-f001:**
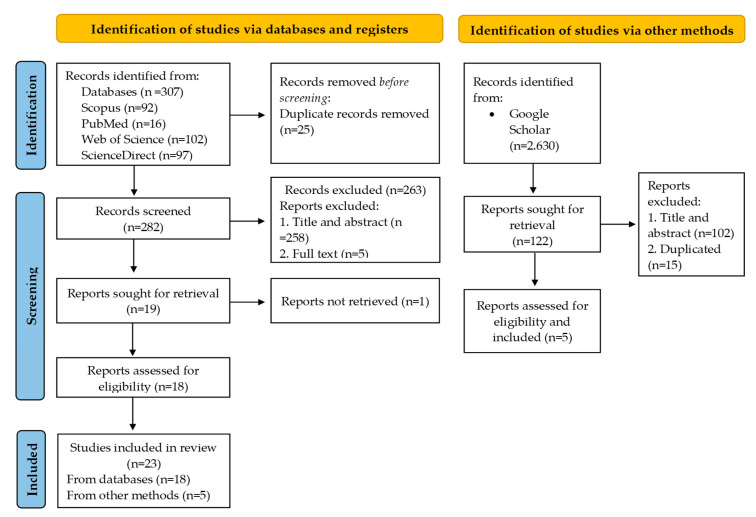
Flow diagram of the systematic review.

**Table 1 sports-14-00003-t001:** Inclusion and exclusion criteria for study selection.

Population	Intervention	Comparison	Outcomes	Study Design
Blind 5-A-side players aiming to train or improve their performance	Physiological variables Physical fitness Competition monitoring GPS Technical and tactical actions Recovery variables Performance levels	Performance levels Blind athletes and sighted athletes Match analysis Competition Effects	Physiological responses (HR, VO_2_max, RPE) Physical Fitness (strength, speed, agility, resistance, balance) Technical–tactical actions (passing, shots on goal, dribbling) Recovery variables (stress, sleep, well-being, muscle pain, fatigue) Competition monitoring (accelerations decelerations total distance)	Cross-sectional study Longitudinal study Observational study Quasi-experimental study Prospective cohort

**Table 2 sports-14-00003-t002:** Quality assessment.

Study (Author, Year)	Design	JBI Tool Applied	Risk of Bias	Main Strengths	Main Limitations
Gamonales et al. [3]	Cross-sectional	Analytical Cross-Sectional Studies	Low–moderate	Real-match data; validated metrics	No confounder control; small sample
Gamonales et al. [5]	Cross-sectional	Analytical Cross-Sectional Studies	Low–moderate	Objective match analysis; validated instrument; high inter-rater reliability	No adjustment for contextual variables
Esatbeyoglu et al. [6]	Cross-sectional	Analytical Cross-Sectional Studies	Low–moderate	Valid measures (IPAQ, Biodex); clear inclusion	No confounder control
Papadopoulos et al. [7]	Cross-sectional	Analytical Cross-Sectional Studies	Moderate	Laboratory VO_2_max test; validated sensors	No confounder control; small sample
Campos et al. [8]	Quasi-experimental (16 weeks pre–post)	Quasi-Experimental Studies	Low–moderate	Standardized training; reliable tests	No control group
Gamonales et al. [9]	Observational study	Studies Reporting Reliability and Validity of Measurement Instruments	Low–moderate	Excellent inter-rater reliability (κ > 0.90); validated instrument	No confounder control
Gamonales et al. [10]	Cross-sectional	Analytical Cross-Sectional Studies	Low–moderate	Objective observation; valid coding (IOLF5C)	No confounder control
Finocchietti et al. [11]	Cross-sectional	Analytical Cross-Sectional Studies	Low–moderate	Valid IMU system; clear procedures	Small sample
Li et al. [20]	Cross-sectional	Analytical Cross-Sectional Studies	Low–moderate	Valid PSQI instrument; complete data	Self-report bias
Li et al. [21]	Prospective cohort (5 months)	Cohort Studies	Low–moderate	Valid tools (ABQ, PSQI); advanced Bayesian analysis; complete follow-up	No confounder control; small sample
Campos et al. [33]	Quasi-experimental (14-week pre–post)	Quasi-Experimental Studies	Low–moderate	Valid physiological measures; complete follow-up	Small sample; no control group
Oliveira et al. [34]	Cross-sectional	Analytical Cross-Sectional Studies	Moderate	Valid fitness protocols; national athletes	No confounder control Small sample (*n* = 5)
Alves et al. [35]	Cross-sectional	Analytical Cross-Sectional Studies	Low–moderate	Objective measures (ergospirometry, torque)	Small sample; no adjustment
Campos et al. [36]	Cross-sectional	Analytical Cross-Sectional Studies	Low–moderate	Standardized isokinetic testing	No covariate control
Nascimento et al. [37]	Cross-sectional	Analytical Cross-Sectional Studies	Low–moderate	Valid force platform; comparison groups	No adjustment for training load
Weiler et al. [38]	Prospective cohort (3 years)	Cohort Studies	Low–moderate	Clear injury definitions; medical verification; long follow-up	No adjusted analysis for confounders; small sample
Ramirez et al. [39]	Cross-sectional	Analytical Cross-Sectional Studies	Moderate	Reliable field tests	No confounder control
Sancio et al. [40]	Cross-sectional	Analytical Cross-Sectional Studies	Moderate	ISAK-certified measurements; objective timing	Very small sample (*n* = 8); descriptive analysis only
Gamonales et al. [41]	Cross-sectional	Analytical Cross-Sectional Studies	Low–moderate	Large data set; validated coding (IOLF5C)	No contextual adjustment
Gamonales et al. [42]	Cross-sectional	Analytical Cross-Sectional Studies	Low–moderate	Valid notational design; strong reliability	No confounder control
Pennell et al. [43]	Cross-sectional	Studies Reporting Reliability and Validity of Measurement Instruments	Low	Strong psychometric analysis (α = 0.81, ω = 0.85) and validity analyses	Small, convenience sample
Monma et al. [44]	Cross-sectional	Analytical Cross-Sectional Studies	Low	Multivariate logistic regression; large sample	Minor self-report bias
Gomes et al. [45]	Quasi-experimental (2-group pre–post)	Quasi-Experimental Studies	Low	strong internal validity, control group, reliable outcomes	Small sample

**Table 3 sports-14-00003-t003:** Classification of methodological procedures in studies according to physical and physiological variables.

Study	Population (*n*, Age, Level, Visual Category)	Intervention/Context	Comparator	Variables/Physiological and Physical	Main Results (Measures, SD, CI if Available)
Gamonales et al. [3]	*n* = 28, (Spain, Italy, Andalusia, Czech Republic), 30.97 ± 11.51 yrs	Official matches International Tournament Phase and result	Tournament Phase and result	Distance, speed, acceleration, loads	Official matches International Changes according to phase (more demands in knockout); variations according to win/lose
Esatbeyoğlu et al. [6]	*n* = 12, B1 soccer players, >18 yrs	Anthropometric tests, PA, balance	Gender and VI	BMI, %fat, fat mass, lean mass, postural balance	VI men > BMI than VI women; women > %fat; postural differences EO vs. foam
Papadopoulos et al. [7]	*n* = 12, players with VI vs. sedentary VI, 27.0 yrs	Official matches International	Blind soccer tournament vs. sedentary	VO_2_max, total distance, speed, HR, anthropometry	Median distance 1820 m (993 m at P50); median speed 2.03 km/h; maximum speed 8.35 km/h; median HR 161 bpm; athletes with ↓fat and ↑VO_2_ vs. sedentary
Campos et al. [8]	*n* = 6, Brazil national team, 27.33 ± 5.5 yrs	Pre vs. post training	Pre vs. Post	Body mass, BMI, % fat, free mass, VO_2_peak, anaerobic power	VO_2_peak: 44.7 → 50.3 mL/kg/min (*p* < 0.05); ↑laps Shuttle Run; ↑anaerobic power; ↓fatigue index
Gamonales et al. [9]	*n* = 50, (Spain, Italy, Andalusia, Czech Republic) 5, 30.86 ± 11.2 yrs	Official matches	Comparisons by result, age, BMI	Accelerations, distance, HRavg, speed	↑Accelerations/min in losers; ↑explosive distance and 21–24 km/h in goalscorers; ↑HRavg and AccMax in younger players
Campos et al. [33]	*n* = 7, Brazilian national team players (B1), 24.7 ± 5.9 yrs	Assessment before/after 14 weeks of training	Pre vs. Post	VO_2_max, HR, ventilatory thresholds, anaerobic power, agility, explosive strength	Significant improvements in VO_2_ and %HRmax at respiratory compensation point; trend towards ↑POpeak, POm, POmin; no changes in FI
Lameira Oliveira et al. [34]	*n* = 5, blind athletes (B1), competitive, Brazil, 32.6 ± 8.0 yrs	Anthropometric assessment + 20 m Shuttle Run	–	Body mass, height, BMI, skinfolds, estimated VO_2_max	BMI 25.1 ± 5.4 (kg/m^2^) within normative range; VO_2_max reported 36.3 ± 4.7 mL^−1^ kg^−1^ min
Alves et al. [35]	*n* = 12, Brazilian Paralympic team (B1), 25.8 ± 5.6 yrs	Cardiorespiratory tests + muscle power	–	VO_2_peak, VO_2_VT1, VO_2_VT2, HRmax, speeds, muscle power, I/Q ratio	VO_2_peak: 51.8 ± 5.8 mL/kg/min; Vmax: 17.1 ± 1.4 km/h; HRVT1: 167.2 ± 10.2 bpm; VO_2_VT2: 48.6 ± 5.7 mL/kg/min
Campos et al. [36]	*n* = 11, pivot/wings/fixed, Brazil national team, 25.9 ± 3.6 yrs	Isokinetic assessment (knee flexors/extensors)	Playing positions (pivot/wings/fixed)	Body mass, height, BMI, %fat, torque, power, peak	PT angle flexors/extensors at 60–300°/s; differences DL vs. NDL (~6%); wings lighter
Nascimento et al. [37]	*n* = 39, B1 vs. B2/B3, 5-A-side soccer, 25.0 ± 5.3 yrs	Postural balance assessment	Blind soccer and grade VI (B2/B3)	Elliptical area, displacement	Speed B1 = smaller displacement area than B2/B3; Footballers = ↓area and speed vs. judokas
Weiler et al. [38]	*n* = 13 blind soccer England, 26.8 ± 4.6 yrs	3 seasons, England	Blind soccer vs. CP	Anthropometry, incidence, severity of injuries	Injury incidence 11.9× higher in matches than training; 73% lower limb injuries; ligaments more common in blind people
Ramirez Roman et al. [39]	*n* = 10 (5 hearing, five visual), Colombia, 28.6 yrs	Jump tests, flexibility, LESS	Disability groups	Vertical/horizontal jump, quadriceps and hamstring flexibility, injury risk	Jumps ≈185 cm visual; flexibility without difference; Higher risk of injury in the left ventricle (LESS test *p* = 0.008)
Sancio et al. [40]	*n* = 8 players from the Argentine national team, 26.8 ± 6.5 years	Old Ball speed tests	Positions (def, mid, for)	Somatotype, % fat, muscle mass, ball speed	Forwards ↑ball speed (4.5 ± 0.22 m/s, max 4.7); correlation r = 0.85 Skeletal index

Note: yrs: years; VI: visual impairment; def: defenders; mid: midfielders; for: forward; m/s: meters over second; max: maximum; BMI: body mass index; m: meters; B1: with residual vision up to 5% in the best eye; HRmax: maximum heart rate; POpeak: máximum power; POm: medium power; POmin: minimum power; FI: fatigue index; I/Q: ischiotibial/quadriceps; VO_2_: O_2_ uptake; VT1: ventilatorythreshold 1; VT2: ventilatorythreshold 2; mL/kg/min: milliliter/kilogram/minute; HRavg: Average Heart Rate; AccMax: maximum acceleration; VO_2_peak: peak oxygen uptake; VO_2_max: maximum oxygen uptake; HR: hear rate; lpm: beats per minute; km/h: kilometers per hour; PA: physical activity; DL: dominant limb; NDL: non-dominant limb; PT: peak torque.

**Table 4 sports-14-00003-t004:** Classification of methodological procedures in studies according to technical and tactical variables.

Study	No Participants/Actions	Age	Population/Sport Level	Method/Instrument	Technical Actions	Technical Indicators	Tactical Indicators	Main Outcome Measures
Gamonales et al. [5]	1497 shots (34 games)—B1	DI	World Cup 2014 official competition	Video analysis + IOLF5C	Shot (type, foot, zone, rebound)	% success, initial zone, contact	Phase, minute, situation, score	Success ↑ if started and finished in the offensive zone; more effective with instep/toe; peaks at 5–10’ and 30–35’.
Gamonales et al. [9]	Experts = 12 (validation)—B1	DI	Amateur, Italy.	Expert judgment, Aiken’s V, Cronbach’s α, κ	Definition of technical variables	Category validation	—	Aiken’s V > 0.875; α = 0.89; high κ.
Gamonales et al. [10]	~424 shots—B1	DI	WGP 2021 (Spain, Thailand, France, Japan, Argentina). International competition	Video + IOLF5C	Control + shot, foot, contact	% with opposition, spatial distribution	Opposition influence, starting zone	Success conditioned by game state, opposition, and striking zone; χ^2^ significant
Finocchietti et al. [11]	*n* = 6 B1	25–38 yrs.	Trained vs. control players. Laboratory/specific actions	Motion capture, EMG, kinematic analysis	Sprint with the ball, turn, strike	Maximum speed, trunk angles	Compensation strategies in postural control	VI < speed and turn; ↑ trunk flexion and compensatory movements; significant differences *p* < 0.05.
Gamonales et al. [41]	730 shots across 18 Paralympic Games matches, B1	DI	Paralympics 2016. Mundial competition	Video + IOLF5C (κ ≈ 0.95)	Type of shot, foot, rebound	% effectiveness by zone, contact.	Context: phase and score.	Initial zone and type of contact predict effectiveness; significant logistic regression analysis.
Gamonales et al. [42]	~2227 shots—B1.	DI	Matches of the 2014 World Cup FA5 (*n* = 34) and the 2016 Paralympic Games (*n* = 18). Different championships compared	Video + IOLF5C	Control + shot, dribble + shot, striking foot	% effectiveness by championship.	Phase and match result.	Significant variations between championships; shot frequency linked to fatigue and phase.
Pennel et al. [43]	*n* = 57, B1	9–18 yrs	Formative, amateur sample, USA	Field tests. Battery of tests: dribbling, passing, shooting	Specific technical tests	Reliability (α, ICC), convergent validity	—	α ≈ 0.81; moderate correlations with shot speed; preliminary validation.

Note: DI: Does not inform; EMG: electromyography; WGP: World Grand Prix; FA5: Football 5-A-side.

**Table 5 sports-14-00003-t005:** Classification of methodological procedures in studies according to recovery variables.

Study	Population (*n*, Age, Level, Visual Category)	Recovery Variables Measured	Instruments/Timing (Measurement Points)	Main Results (Means, % or Key Findings)
Li et al. [20]	*n* = 60 blind athletes (B1); mean age 22.8 ± 4.5 years; training ≈ 19.9 ± 8.0 h/week	Overall sleep quality and components (PSQI: subjective, latency, duration, efficiency, disturbances, medication use, daytime dysfunction).	PSQI administered by survey; comparison with a secondary dataset of athletes without disabilities.	Total PSQI 4.42 ± 2.70; 26.7% classified as “poor sleepers”. Significant differences in components (latency, duration, efficiency, dysfunction) as a function of training volume (better sleepers with higher volume). No medication use reported
Li et al. [21]	*n* = 10 (blind national selection from China).	Sleep quality (PSQI or other self-report sleep measure) and burnout (sport burnout instruments).	Monthly interviews/surveys (month 1 to month 5); dynamic analysis (p-technique/Bayesian for short series).	Lagged effect: burnout predicts worsening sleep quality in subsequent time steps; sleep does not consistently predict burnout. Robust finding in dynamic analysis despite small *n*.
Monma et al. [44]	A survey of *n* = 99 visually impaired athletes was conducted; 81 responses were analyzed (72.8% male); mean age 32.5 ± 12.0 years.	Sleep disorders (prevalence), sleep habits, and risk factors (stress from interpersonal relationships, schedules)	PSQI; cutoff point for disorder: PSQI ≥ 5.5; analysis by multivariate logistic regression	26/81 (32.1%) presented with sleep disorder (PSQI ≥ 5.5). Associated independent factors: interpersonal relationship stress and late wake-up time.
Gomes et al. [45]	*n* = 8 visually impaired athletes vs. *n* = 15 blind (non-visually impaired) athletes (males).	Blood markers: biomarkers of oxidative stress, antioxidant capacity, CK (muscle damage marker), lactate, liver enzymes, cortisol, testosterone	Blood samples were taken before and after maximal exercise testing; biochemical analyses (TBARS, CK, GGT, ALT/AST, cortisol)	Non-visual players showed greater aerobic capacity (*p* < 0.05). Lactate increased fourfold post-test. CK, GGT, and oxidative and antioxidant biomarkers did not change significantly. Cortisol increased post-test; ALT/AST increased only in the non-visual players. Conclusion: Blind players showed less cellular damage after the test than the sighted group

Note: PSQI: Pittsburgh Sleep Quality Index; CK: creatine kinasa; GGT: gamma-glutamyltransferase; ALT: alanine aminotransferase; AST: Aspartate Aminotransferase; TBARS: thiobarbituric acid reactive substances; h: hour.

## Data Availability

No new data were created or analyzed in this study. Data sharing is not applicable to this article.

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
