# Peer review of "Identification of Performance Variables in Blind 5-A-Side Football: Physical Fitness, Physiological Responses, Technical–Tactical Actions and Recovery Variables: A Systematic Review"

_sports, 2026, doi:10.3390/sports14010003_

Round 1

Reviewer 1 Report

Comments and Suggestions for Authors

However, despite the breadth of the review, I find that the manuscript currently functions more as a descriptive compilation of findings than as a critically integrative systematic review. Several conceptual, methodological, and interpretive issues limit its scientific contribution and should be addressed before publication.

  1. Conceptual framing and theoretical integration

While the Introduction provides a clear description of blind 5-a-side football and justifies the need for a review, I did not find a clearly articulated conceptual framework guiding the synthesis. Performance variables are treated as parallel categories rather than as interacting components within an integrated performance model specific to visually impaired football. As a result, the review risks presenting an inventory of variables rather than explaining how and why these variables matter for performance under sensory constraints.

What should be corrected:

The authors should explicitly articulate a conceptual or theoretical framework that explains how physical, technical–tactical, and recovery variables interact in blind football (e.g., through constraint-based, ecological, or systems-oriented perspectives).

My expectation:

I expect the revised manuscript to move beyond categorization and to provide an integrative model or narrative that clarifies the relationships between performance domains and the unique demands imposed by visual impairment.

  1. Methodological consistency and inclusion criteria

I appreciate that the review follows PRISMA guidelines and applies a PICOS strategy. However, I noted an important inconsistency between the stated inclusion criteria and the studies actually synthesized. The Methods section indicates that secondary literature (systematic reviews, narrative reviews, bibliometric analyses) was eligible, yet the Results section synthesizes only primary quantitative studies. This discrepancy undermines transparency and reproducibility.

What should be corrected:

The authors should either (a) revise the inclusion criteria to reflect that only primary quantitative studies were included in the synthesis, or (b) explicitly integrate and discuss the secondary literature if it was indeed eligible.

My expectation:

I expect full alignment between eligibility criteria, study selection, and synthesis, so that readers can clearly understand what evidence base the conclusions are drawn from.

  1. Risk of bias assessment and strength of evidence

The use of JBI tools for quality assessment is appropriate, and I appreciate that risk-of-bias evaluation was conducted. However, I find a mismatch between the reported prevalence of low-to-moderate risk of bias and the acknowledged methodological limitations of the included studies, such as small sample sizes, lack of control groups, and limited adjustment for confounders. These limitations suggest that the overall strength of evidence is modest.

What should be corrected:

The authors should more clearly link the quality assessment results to the strength of their conclusions and explicitly acknowledge how methodological weaknesses constrain inference.

My expectation:

I expect the interpretation of findings and the practical implications to be consistently calibrated to the actual quality and design of the underlying studies.

  1. Depth of synthesis and critical comparison

Across the physical, technical–tactical, and recovery domains, the Results and Discussion sections remain largely descriptive. While the authors accurately report variables such as VO₂max, accelerations, technical actions, and sleep parameters, I found limited critical comparison across studies. Differences in study design, competitive level, visual classification, and measurement tools are mentioned but not systematically integrated into the interpretation.

What should be corrected:

The authors should engage in more comparative and critical synthesis, explicitly discussing how methodological heterogeneity, population differences, and contextual factors influence the consistency and applicability of findings.

My expectation:

I expect the revised Discussion to highlight convergence, divergence, and uncertainty across studies, rather than presenting findings as broadly comparable.

  1. Interpretation and practical implications

The manuscript offers several practical recommendations for coaches and practitioners, which I view as a strength. However, given that most included studies are cross-sectional or observational, some interpretations and applied claims appear stronger than warranted. In particular, suggestions related to training optimization and performance enhancement are not always sufficiently qualified.

What should be corrected:

The authors should adopt a more cautious and conditional tone when translating findings into practice, clearly distinguishing between evidence-supported conclusions and promising but untested implications.

My expectation:

I expect practical recommendations to be explicitly framed as provisional and context-dependent, in line with the exploratory nature of the evidence base.

  1. Conclusions and alignment with limitations

The Limitations section is relatively thorough and appropriately acknowledges issues such as heterogeneity, population bias toward elite male athletes, and limited generalizability. However, I did not find these limitations fully reflected in the Conclusions, which remain somewhat optimistic in tone.

What should be corrected:

The Conclusions should more directly reflect the limitations discussed and avoid implying a level of certainty that exceeds the available evidence.

My expectation:

I expect stronger coherence between the acknowledged limitations and the final claims of the manuscript.

Author Response

Letter to reviewer 1

Dear Reviewer,

Thank you for reviewing our manuscript. As a research group we value your effort and input.

We have followed your suggestions point by point to improve the manuscript quality, according to our possibilities. The changes have been made in the full text using the red color so that you can see them.

Thanks for your time. Once again, we thank you for your valuable contributions, which have helped to strengthen the document.

  1. Conceptual framing and theoretical integration

Comments 1. While the Introduction provides a clear description of blind 5-a-side football and justifies the need for a review, I did not find a clearly articulated conceptual framework guiding the synthesis. Performance variables are treated as parallel categories rather than as interacting components within an integrated performance model specific to visually impaired football. As a result, the review risks presenting an inventory of variables rather than explaining how and why these variables matter for performance under sensory constraints.

Response 1. Thank you very much for your comments. The reviewer's comments were accepted.

The introduction was restructured to give it a more inclusive narrative model. The following sections were added:

Technical and tactical variables are considered fundamental for adapting to the specific context and situation of the competition, which has proven to be an indicator of success, such as shots on goal [10]. Blind 5-a-side football has evolved in its technical and tactical demands, moving from a game based on dribbling and shooting on goals to a more combinative game through the dynamics of passing, controlling, and dribbling to achieve the objective of the game by kicking the ball toward the goal [10]. This precision is favored when there is greater balance [11], a quality that directly affects technical execution in the sport [12]. Likewise, 5-a-side soccer for the blind is a team sport in which physical contact is constant due to visual impairment, and the game requires high-intensity technical skills, integrating physiological systems with the main technical-tactical characteristics [13] and the expression of physical variables for passing, dribbling, and shooting. In this sport, players express different technical and tactical characteristics, as well as physical, physiological, and morphological traits that allow them to adapt to unpredictable situations [14, 15], which together constitute determinants of performance [16]. Therefore, performance in blind 5-a-side soccer is considered a multifactorial, complex, and dynamic process [17,18,19] that requires the integration of physical abilities, physiological processes, and recovery processes that enhance technical skills within a framework of tactical understanding of the game.

Athletic performance in blind 5-a-side soccer is a complex process that requires the interaction of different abilities, including aerobic-anaerobic systems, muscle strength, and balance, which, together with dribbling speed, shooting accuracy and power, and coordination, are essential for adapting to the demands of competition [23,24,25]. Blind 5-a-side soccer is an intermittent sport with specific requirements associated with high physiological, physical, technical, and tactical demands that optimize player performance in a competitive context with high levels of uncertainty [26]. Therefore, knowing which performance variables have been most studied in scientific literature could help to understand the main characteristics of athletes and the least explored variables with a view to designing an increasing ecological perspective of this sport [27].

  1. Methodological consistency and inclusion criteria

Comments 2. I appreciate that the review follows PRISMA guidelines and applies a PICOS strategy. However, I noted an important inconsistency between the stated inclusion criteria and the studies actually synthesized. The Methods section indicates that secondary literature (systematic reviews, narrative reviews, bibliometric analyses) was eligible, yet the Results section synthesizes only primary quantitative studies. This discrepancy undermines transparency and reproducibility.

Response 2. Thank you very much for your comments. The reviewer's comments were accepted.

The eligibility criteria were adjusted to harmonize the search with the documents found and included in the review. An inclusion criterion was added: explicitly integrate and discuss secondary literature.

The inclusion criterion for review studies was removed and changed to an exclusion criterion, because all the studies included are primary quantitative studies.

  1. Risk of bias assessment and strength of evidence

Comments 3. The use of JBI tools for quality assessment is appropriate, and I appreciate that risk-of-bias evaluation was conducted. However, I find a mismatch between the reported prevalence of low-to-moderate risk of bias and the acknowledged methodological limitations of the included studies, such as small sample sizes, lack of control groups, and limited adjustment for confounders. These limitations suggest that the overall strength of evidence is modest.

Response 3. Thank you very much for your comments. The reviewer's comments were accepted.

The conclusions were restructured to highlight the limitations of the low quality of evidence due to the heterogeneity of the studies, design used, instruments employed, sample sizes, athletic levels, etc. The findings have been analyzed based on the available scientific evidence, so it is suggested that the findings presented in this review be taken with caution.

  1. Depth of synthesis and critical comparison

Comments 4. Across the physical, technical–tactical, and recovery domains, the Results and Discussion sections remain largely descriptive. While the authors accurately report variables such as VO₂max, accelerations, technical actions, and sleep parameters, I found limited critical comparison across studies. Differences in study design, competitive level, visual classification, and measurement tools are mentioned but not systematically integrated into the interpretation.

Response 4. Thank you very much for your comments. The reviewer's comments were accepted.

However, the synthesis of the review was based on the available evidence to discuss the findings in response to the variables that were analyzed in each study. Contextual factors were influential variables that hindered the consistency and applicability of the findings. Therefore, an explanation of the main difficulties and limitations in presenting the findings was added.

The discussion sought to group together the outcome measures from the studies that analyzed these variables. Due to the low number of studies that addressed the same variables, these findings were grouped together. Throughout the manuscript, including the discussion, conclusions, limitations, and practical applications, it is mentioned that the conclusions of this study should be taken with caution due to the methodological robustness (cross-sectional, descriptive studies, small sample sizes) presented in these included studies.

However, it should be noted that the methodological heterogeneity of the included studies, population differences, and contextual factors significantly influence the consistency and applicability of the findings. Therefore, the main variables analyzed in each of the performance indicators examined in this review are presented below.

  1. Interpretation and practical implications

Comments 5. The manuscript offers several practical recommendations for coaches and practitioners, which I view as a strength. However, given that most included studies are cross-sectional or observational, some interpretations and applied claims appear stronger than warranted. In particular, suggestions related to training optimization and performance enhancement are not always sufficiently qualified.

Response 5. Thank you very much for your comments. The reviewer's comments were accepted.

The practical recommendations were formulated considering the limitations of the findings and the needs of athletes, the instruments used, and the context.

The results of this review provide useful information for coaches, physical trainers, and sports scientists. However, it should be noted that these considerations should be generalized with caution, given the exploratory nature of the evidence, the heterogeneity of the samples evaluated and instruments used, and the differences in each context intervened upon. Based on the interpretation of the results obtained from the studies included and the variables analyzed in this review, training programs should focus on developing aerobic capacity, seeking to achieve maximum oxygen consumption (≈50 ml·kg⁻¹·min⁻¹), as well as stimulating explosive strength and postural control. For technical-tactical variables, it is suggested that exercises focus on striking with the instep or toe and in simulated situations of low opposition from offensive areas, which is likely to increase the effectiveness of shots on goal. From a recovery perspective, sleep control, subjective well-being, and pain perception could be preventive indicators of the risk of overtraining and injury. Performance analysis in blind 5-a-side soccer requires comprehensive preparation and the use of tools adapted to the needs of the athletes.

  1. Conclusions and alignment with limitations

Comments 6. The Limitations section is relatively thorough and appropriately acknowledges issues such as heterogeneity, population bias toward elite male athletes, and limited generalizability. However, I did not find these limitations fully reflected in the Conclusions, which remain somewhat optimistic in tone.

Response 6. Thank you very much for your comments. The reviewer's comments were accepted.

The conclusions were reformulated to ensure greater consistency between the limitations and the findings derived from the studies.

Although the available scientific evidence is not consistent, it appears that certain variables related to physical, physiological, technical-tactical, and recovery factors may influence the athletic performance of players who participate in blind 5-a-side soccer. Some of the most representative physical variables appear to be balance and postural control, while cardiorespiratory fitness seems to be one of the physiological variables that influence performance. Among the technical-tactical variables, one of the most representative is accuracy in shooting in offensive areas with minimal opposition.

Scientific output in this discipline remains limited, highlighting the need to promote studies with greater methodological rigor and sample diversity, which makes it difficult to extrapolate these findings to different contexts. Therefore, it is suggested that these findings be reviewed with caution. This review lays the foundation for future research and could serve as input to guide practical applications that seek to understand the athletic performance of blind soccer players.

Thank you for your positive feedback on our research. Your valuable suggestions greatly contributed to the improvement of our work.

Best regards

Reviewer 2 Report

Comments and Suggestions for Authors

General comments

This systematic review addresses an important and under-researched area by synthesising performance variables in blind 5-a-side football. The topic is timely, relevant, and has clear applied value for coaches, sport scientists, and Paralympic practitioners. The authors present a well-structured rationale, provide detailed tables of study characteristics, and include a registered protocol and PRISMA flow diagram, which strengthens the transparency of the review. However, several major issues must be addressed before the manuscript can be considered for publication.

First, adherence to PRISMA 2020 reporting standards is partial: key methodological components and most notably reproducible search strategies, data extraction procedures, synthesis methods, and integration of risk-of-bias assessments into interpretation are insufficiently reported.

Second, the manuscript frequently labels samples as “elite” despite considerable heterogeneity in competitive level, small sample sizes, and inconsistent participant classification across studies. This risks overstating the generalisability of findings.

Third, several claims in the Discussion and Conclusions are stronger than the underlying evidence supports; many included studies are observational, cross-sectional, or descriptive, and therefore cannot justify causal or deterministic interpretations.

Specific

  1. A major issue concerns the manuscript’s repeated use of the term “elite” to describe the sampled populations; eg “elite 5-a-side soccer players” in Table 3 and throughout the Results section . Yet, the studies included differ markedly in competitive standard: some involve national Paralympic squads (e.g., Brazilian national team), others involve regional squads (Spain, Italy, Andalusia, Czech Republic), and several studies contain very small samples (e.g., n=5 or n=6) drawn from training groups rather than verified elite teams. The manuscript therefore risks overstating the competitive caliber represented in the evidence base. You should clarify the classification criteria used by each included study and avoid homogenising all participants as “elite,” unless justified using an accepted framework. Without this clarification, readers may incorrectly infer that the review synthesises performance determinants of world-class Paralympic football, when in fact the evidence mostly reflects heterogeneous, small, convenience samples.

Similarly, some conclusions overstate what the underlying evidence can genuinely support. For example, the manuscript concludes that “elite players have cardiorespiratory values comparable to conventional soccer players” and that certain technical-tactical patterns “determine success” . However, the aerobic-fitness comparisons are generally made against broad population benchmarks rather than matched samples, and technical-tactical findings derive largely from descriptive notational analyses without controlling for situational constraints, opposition strength, or player role. Many included studies do not control for confounders, have moderate-to-high risk of bias, and rely on very small samples (as noted in the JBI assessment on pp. 4–5) . Accordingly, stronger language such as “determine,” “predict,” or “are key determinants” should be replaced with more cautious phrasing such as “are associated with,” “are frequently observed in,” or “may influence.” Tempering claims will strengthen the scientific credibility of the review.

Discuss sample heterogeneity more explicitly. Many studies combine B1 players with B2/B3 or compare them to sighted participants. Yet visual classification profoundly changes the perceptual–motor demands of the sport. This should be presented as a methodological limitation with implications for interpreting “performance determinants.”

Strengthen the synthesis by distinguishing evidence quality. The Discussion tends to group findings regardless of methodological strength. Consider weighting conclusions based on study design quality (e.g., intervention studies vs. small descriptive studies).

Address generalisability. The dataset is overwhelmingly male and heavily centred on South American and European samples, with limited representation from youth, female athletes, and emerging nations. This needs stating more clearly.

Author Response

Letter to reviewer 2

Dear Reviewer,

Thank you for reviewing our manuscript. As a research group we value your effort and input.

We have followed your suggestions point by point to improve the manuscript quality, according to our possibilities. The changes have been made in the full text using the red color so that you can see them.

Thanks for your time. Once again, we thank you for your valuable contributions, which have helped to strengthen the document.

Comments 1. First, adherence to PRISMA 2020 reporting standards is partial: key methodological components and most notably reproducible search strategies, data extraction procedures, synthesis methods, and integration of risk-of-bias assessments into interpretation are insufficiently reported.

Response 1. Thank you very much for your comments. The reviewer's comments were accepted.

All material and methods were thoroughly reviewed, ensuring that each methodological aspect, such as search strategy, data extraction, synthesis method, and bias risk assessment, was sufficiently reported.

Additionally, some inclusion/exclusion criteria were adjusted to strengthen the study.

The eligibility criteria were adjusted to harmonize the search with the documents found and included in the review. An inclusion criterion was added: explicitly integrate and discuss secondary literature.

The inclusion criterion for review studies was removed and changed to an exclusion criterion, because all the studies included are primary quantitative studies. 

Comments 2. Second, the manuscript frequently labels samples as “elite” despite considerable heterogeneity in competitive level, small sample sizes, and inconsistent participant classification across studies. This risks overstating the generalisability of findings.

Response 2. Thank you very much for your comments. The reviewer's comments were accepted.

The use of vocabulary that would generalize the results was reviewed and adjusted so as not to overestimate the population samples analyzed in the study based on the available scientific evidence. The concept of “elite” was removed from every section of the manuscript.

Comments 3. Third, several claims in the Discussion and Conclusions are stronger than the underlying evidence supports; many included studies are observational, cross-sectional, or descriptive, and therefore cannot justify causal or deterministic interpretations.

Response 3. Thank you very much for your comments. The reviewer's comments were accepted.

The reviewer's contributions were considered to avoid assertions in the discussion and conclusions based on the limitations presented by the methodological quality of the available scientific evidence. A paragraph was added to the discussion to comment on the difficulties in the heterogeneity of the studies, and the conclusions were restructured to avoid causal or deterministic interpretations.

The elements of the discussion and conclusions were adjusted to avoid causal or deterministic interpretations, which are not possible due to the heterogeneity of the results and the design of the included studies. 

The sections that were added and/or modified were as follows:

This is the first systematic review to analyze scientific evidence on performance variables in players with visual impairments. In view of this, the outcome measures in this systematic review were those evaluated in at least 3 of the 23 articles. However, it should be noted that the methodological heterogeneity of the included studies, population differences, and contextual factors significantly influence the consistency and applicability of the findings. Therefore, the main variables analyzed in each of the performance indicators examined in this review are presented below.

The results of this review provide useful information for coaches, physical trainers, and sports scientists. However, it should be noted that these considerations should be generalized with caution, given the exploratory nature of the evidence, the heterogeneity of the samples evaluated and instruments used, and the differences in each context intervened upon. Based on the interpretation of the results obtained from the studies included and the variables analyzed in this review, training programs should focus on developing aerobic capacity, seeking to achieve maximum oxygen consumption (≈50 ml·kg⁻¹·min⁻¹), as well as stimulating explosive strength and postural control. For technical-tactical variables, it is suggested that exercises focus on striking with the instep or toe and in simulated situations of low opposition from offensive areas, which is likely to increase the effectiveness of shots on goal. From a recovery perspective, sleep control, subjective well-being, and pain perception could be preventive indicators of the risk of overtraining and injury. Performance analysis in blind 5-a-side soccer requires comprehensive preparation and the use of tools adapted to the needs of the athletes.

Although the available scientific evidence is not consistent, it appears that certain variables related to physical, physiological, technical-tactical, and recovery factors may influence the athletic performance of players who participate in blind 5-a-side soccer. Some of the most representative physical variables appear to be balance and postural control, while cardiorespiratory fitness seems to be one of the physiological variables that influence performance. Among the technical-tactical variables, one of the most representative is accuracy in shooting in offensive areas with minimal opposition.

Scientific output in this discipline remains limited, highlighting the need to promote studies with greater methodological rigor and sample diversity, which makes it difficult to extrapolate these findings to different contexts. Therefore, it is suggested that these findings be reviewed with caution. This review lays the foundation for future research and could serve as input to guide practical applications that seek to understand the athletic performance of blind soccer players.

The conclusions were restructured to highlight the limitations of the low quality of evidence due to the heterogeneity of the studies, design used, instruments employed, sample sizes, athletic levels, etc. The findings have been analyzed based on the available scientific evidence, so it is suggested that the findings presented in this review be taken with caution.

Comments 4.  A major issue concerns the manuscript’s repeated use of the term “elite” to describe the sampled populations; eg “elite 5-a-side soccer players” in Table 3 and throughout the Results section . Yet, the studies included differ markedly in competitive standard: some involve national Paralympic squads (e.g., Brazilian national team), others involve regional squads (Spain, Italy, Andalusia, Czech Republic), and several studies contain very small samples (e.g., n=5 or n=6) drawn from training groups rather than verified elite teams. The manuscript therefore risks overstating the competitive caliber represented in the evidence base. You should clarify the classification criteria used by each included study and avoid homogenising all participants as “elite,” unless justified using an accepted framework. Without this clarification, readers may incorrectly infer that the review synthesises performance determinants of world-class Paralympic football, when in fact the evidence mostly reflects heterogeneous, small, convenience samples.

Response 4. Thank you very much for your comments. The reviewer's comments were accepted.

The use of vocabulary that would generalize the results was reviewed and adjusted so as not to overestimate the population samples analyzed in the study based on the available scientific evidence. The concept of “elite” was removed from every section of the manuscript.

However, the synthesis of the review was based on the available evidence to discuss the findings in response to the variables that were analyzed in each study. Contextual factors were influential variables that hindered the consistency and applicability of the findings. Therefore, an explanation of the main difficulties and limitations in presenting the findings was added. 

Comments 5. Similarly, some conclusions overstate what the underlying evidence can genuinely support. For example, the manuscript concludes that “elite players have cardiorespiratory values comparable to conventional soccer players” and that certain technical-tactical patterns “determine success” . However, the aerobic-fitness comparisons are generally made against broad population benchmarks rather than matched samples, and technical-tactical findings derive largely from descriptive notational analyses without controlling for situational constraints, opposition strength, or player role. Many included studies do not control for confounders, have moderate-to-high risk of bias, and rely on very small samples (as noted in the JBI assessment on pp. 4–5) . Accordingly, stronger language such as “determine,” “predict,” or “are key determinants” should be replaced with more cautious phrasing such as “are associated with,” “are frequently observed in,” or “may influence.” Tempering claims will strengthen the scientific credibility of the review.

Response 5. Thank you very much for your comments. The reviewer's comments were accepted.

The conclusions were restructured to highlight the limitations of the low quality of evidence due to the heterogeneity of the studies, design used, instruments employed, sample sizes, athletic levels, etc. The findings have been analyzed based on the available scientific evidence, so it is suggested that the findings presented in this review be taken with caution.

Comments 6. Discuss sample heterogeneity more explicitly. Many studies combine B1 players with B2/B3 or compare them to sighted participants. Yet visual classification profoundly changes the perceptual–motor demands of the sport. This should be presented as a methodological limitation with implications for interpreting “performance determinants.”

Response 6. Thank you very much for your comments. The reviewer's comments were accepted.

Finally, a significant limitation is that some studies combine B1 players with B2/B3 players and sighted participants, which introduces a significant methodological bias in the interpretation of results when it comes to understanding the determinants of performance in blind 5-a-side soccer players.

In addition, there are other limitations that are important to mention.

Notably, the authors of studies whose full texts were unavailable were not contacted, leading to the exclusion of potentially relevant research and potentially introducing availability bias. Furthermore, the methodological heterogeneity of the included studies, both in terms of sample size and experimental design (cross-sectional, observational, or longitudinal), prevents direct comparisons or quantitative meta-analyses. In addition, most research has focused on high-performance male populations, limiting the generalizability of the findings to visually impaired soccer players at the training level or to female players. Likewise, the limited availability of databases and the scarcity of publications in other languages may have restricted the identification of relevant studies.

The lack of experimental control over variables such as visual category (B1–B3), playing position, or stage of the season introduces potential biases in the interpretation of performance results. Another significant limitation is the lack of information on playing positions and their differences in performance variables, which prevented robust comparisons between tactical roles within the team. Overall, the heterogeneity in designs, sample sizes, and methodologies across the included studies limits quantitative comparisons and meta-analyses, thereby restricting the generalizability of the findings.

Comments 7. Strengthen the synthesis by distinguishing evidence quality. The Discussion tends to group findings regardless of methodological strength. Consider weighting conclusions based on study design quality (e.g., intervention studies vs. small descriptive studies).

Response 7. Thank you very much for your comments

However, the synthesis of the review was based on the available evidence to discuss the findings in response to the variables that were analyzed in each study. Contextual factors were influential variables that hindered the consistency and applicability of the findings. Therefore, an explanation of the main difficulties and limitations in presenting the findings was added.

The discussion sought to group together the outcome measures from the studies that analyzed these variables. Due to the low number of studies that addressed the same variables, these findings were grouped together. Throughout the manuscript, including the discussion, conclusions, limitations, and practical applications, it is mentioned that the conclusions of this study should be taken with caution due to the methodological robustness (cross-sectional, descriptive studies, small sample sizes) presented in these included studies.

However, it should be noted that the methodological heterogeneity of the included studies, population differences, and contextual factors significantly influence the consistency and applicability of the findings. Therefore, the main variables analyzed in each of the performance indicators examined in this review are presented below.

Comments 8. Address generalisability. The dataset is overwhelmingly male and heavily centred on South American and European samples, with limited representation from youth, female athletes, and emerging nations. This needs stating more clearly.

Response 8. Thank you very much for your comments

Essentially, all of the studies found were on male athletes, focusing on South American and European samples. This is an opportunity for further research.

Future research should expand the study of technical-tactical variables using tracking technologies (GPS, accelerometry, computer vision, and three-dimensional kinematic analysis), as well as the integration of psychophysiological measures (stress, well-being, and sleep) in competitive contexts. It would also be relevant to explore the relationship between anthropometric indicators and neuromuscular adaptations to establish performance profiles. Longitudinal studies should also be developed to evaluate the impact of specific training programs on the effectiveness of technical-tactical actions and on injury prevention. Finally, the inclusion of women and young athletes with visual impairments should be promoted to expand the evidence base on sex, age, countries and competitive-level diversity.

Thank you for your positive feedback on our research. Your valuable suggestions greatly contributed to the improvement of our work.

Best regards

Round 2

Reviewer 2 Report

Comments and Suggestions for Authors

The article has improved. 

We need

a) revisit the term elite - it appears in the Tables and so provide an explanation - say this is how the authors labelled the sample, and then engage the reader in a discussion on the challenges of labelling a sample elite. 

b) Risk of bias - this is presented as a limitation but in the methods it says it was undertaken and so please fix this inconsistency. 

Author Response

Letter to reviewer 2

Dear Reviewer,

Thank you for reviewing our manuscript. As a research group we value your effort and input.

We have followed your suggestions point by point to improve the manuscript quality, according to our possibilities. The changes have been made in the full text using the red color so that you can see them.

Thanks for your time. Once again, we thank you for your valuable contributions, which have helped to strengthen the document.

Comments 1. a) revisit the term elite - it appears in the Tables and so provide an explanation - say this is how the authors labelled the sample, and then engage the reader in a discussion on the challenges of labelling a sample elite. 

Response 1. Thank you very much for your comments. The reviewer's comments were accepted.

The concept of “elite” was reviewed in the tables. This classification was corrected and the concept was removed from all tables. The tables also include a description of the context of the sample and the level of sport, which is why international matches and Olympic games appear. The aim is to avoid labels that could lead to ambiguity in the level of the sample and to make it easier for readers to understand the results and analysis derived from this review.

Comments 2. b) Risk of bias - this is presented as a limitation but in the methods it says it was undertaken and so please fix this inconsistency. 

Response 2. Thank you very much for your comments. The reviewer's comments were accepted.

This inconsistency in the assessment of risk of bias was corrected. This section was removed from the limitations section.

Thank you for your positive comments on our research. Your valuable suggestions in this second phase of review have greatly contributed to the improvement of our work.

Best regards